# Investigation of the reasons for delayed presentation in proliferative diabetic retinopathy patients

**Meng Zhao[1], Aman Chandra[2,3‡], Lin Liu[1‡], Lin Zhang[4‡], Jun Xu[1‡], Jipeng Li[1]***

**1** Ophthalmology, Beijing Tongren Eye Center, Beijing Key Laboratory of Ophthalmology and Visual Science, Beijing Tongren Hospiospital, Dongcheng District, Beijing, 100730, China, **2** Mid & South Essex NHS Foundation Trust (Southend University Hospital) Prittlewell Chase Essex SS00RY, Southend-on-Sea, United Kingdom, **3** Anglia Ruskin University, Cambridge, United Kingdom, **4** Endocrinology, Beijing Tongren Hospital, Capital Medical University, Dongcheng District, Beijing, 100730, China

☯ These authors contributed equally to this work.
‡ These authors also contributed equally to this work.
* jipeng2004@sina.com

**Data Availability Statement:** The original data can be obtained at the zhao, meng (2023), "questionnaire for PDR delayed presentation", Mendeley Data, V1, doi: 10.17632/z956gdk4f7.1.

## Abstract

### Aim

To investigate reasons for delayed presentation in patients with proliferative diabetic retinopathy (PDR).

### Methods

A questionnaire was designed to investigate consecutive PDR patients with delayed presentation who visited our center between January 2021 and December 2021. The questionnaire was divided into four sections: knowledge regarding diabetic retinopathy (DR), attitude toward DR treatment, difficulties adhering to follow-up plans, and medical care. The systemic disease status and severity of DR were recorded. Logistic analysis was undertaken to investigate DR treatment refusal and delay factors.

### Results

A total of 157 patients were included in this study, with an average age of 50.0 ± 11.6 years. The median glycated hemoglobin level (HbA1c) was 7.8% (IQR 2.5%). Among the 157 eyes, most required vitrectomy intervention (144, 91.7%); 17 developed neovascular glaucoma (NVG), while only 13 required additional photocoagulation. Among the 36 patients with undiagnosed DM, the reason for delayed DR presentation was a lack of awareness of DM status among these patients (36 cases, 100.0%). Most of the patients with a known history of DM exhibited inadequate DR knowledge (29, 24.0%), believed their good visual acuity did not require DR screening (98, 81.0%), and had poorly controlled diabetes (113, 93.3%). Factors related to refusing DR treatment were patients with an inability to receive regular diabetes treatment in internal medicine clinics (OR 6.78, 95% CI 1.73–26.59, p = 0.006), patients who could not tolerate discomfort during ophthalmic examination and treatment (OR 15.15, 95% CI 2.70–83.33, p<0.001), and patients who did not have any retinal

**Funding:** The author(s) received no specific funding for this work.

**Competing interests:** The authors have declared that no competing interests exist.

**Abbreviations:** AIC, Akaike information criterion; CKD, chronic kidney disease; DR, diabetic retinopathy; HbA1c, glycated hemoglobin level; HTN, hypertension; IV, intravitreal injection; NVI, neovascularization of the iris; NVG, neovascular glaucoma; PDR, proliferative diabetic retinopathy; PRP, pan-retinal photocoagulation; VEGF, vascular endothelial growth factor; VH, vitreous hemorrhage.

abnormalities detected and were not informed about the need for regular screening (OR 2.05, 95% CI 1.36–3.09, p<0.001).

## Conclusions

This study investigated the factors contributing to delayed presentation among patients with PDR. Many individuals in the delayed population were found to have undiagnosed DM. Among patients already aware of their DM status, reasons for delay included insufficient knowledge about DR, negative attitudes toward screening and treatment, and difficulties seeking medical care in real-life situations. Furthermore, there needed to be more improvements in the detection, treatment, and follow-up of DR by internal medicine practitioners and ophthalmologists.

## Introduction

Diabetic retinopathy (DR), a consequential complication of diabetes, stands as the predominant cause of blindness among the working-age population [1]. Early detection and prompt treatment of PDR play a crucial role in reducing the risk of diabetes-related blindness [2,3].

In countries implementing comprehensive screening programs for DR, the annual incidence of blindness resulting from DR has declined [4,5]. However, despite the availability of national DR screening in some countries, many patients still do not receive timely treatment. This issue is not exclusive to specific regions, as highlighted by a national survey that revealed alarming statistics. A survey found that 70.1% of DR patients identified through fundus photography were unaware of their DR condition, while 23.1% of patients with known diabetes were unaware of their DM status [6].

In China, the lack of a national DR screening program and data on delayed DR presentation further exacerbates the problem. However, the increasing number of cases of DR-related blindness in the country indicates a pressing need for prompt action. Between 1999 and 2019, the annual incidence rate of DR-related blindness in China skyrocketed from 0.06 to 0.23 cases per million population, and the rate of visual impairment rose from 0.3 to 0.86 cases per million population [7]. Examining regional data sheds further light on the extent of the issue. For instance, a survey conducted among elderly residents in Beijing communities discovered that 87% of patients with DR were unaware of their condition. Moreover, only 28% of diabetes patients had undergone DR screening within the past year [8]. Similarly, an epidemiological survey conducted in rural northern China revealed that 12.1% of the diabetes patients had vision-threatening DR but had not received the necessary treatment [9]. These regional findings emphasize a significant delay in diagnosing and treating PDR within the diabetic population in China.

Emerging research [10,11], highlights a complex relationship between inflammation and various diabetic complications, including diabetic neuropathy [12], DR [13], and diabetic nephropathy [14]. This growing body of evidence suggests that the inflammatory characteristics of Type 2 DM may contribute to the progression of DR [10–14]. Specifically, studies have shown that individuals with poorly controlled diabetes, characterized by heightened inflammation [10,11], exhibit a higher risk of both faster DR progression [13] and delayed presentation due to the poor compliance [15]. Thus, early detection of diabetes and its complications could improve effective management of the disease.

Various factors contribute to the delay in the diagnosis and treatment of DR. In countries where national DR screening programs have been implemented, it has been shown that reasons for the patient delay or nonparticipation in screening include insufficient social support [15] and poor patient compliance [16], inadequate awareness of DR [15], lack of DR-related visual symptoms [17,18], limited opportunities for dilated fundus examination [19], restrictions on seeking medical care due to systemic diseases [20], economic factors [16], racial differences [21] pain during treatment [22], and inability to achieve expected visual outcomes [17,18].

The prevalence of diabetes among Chinese patients has increased annually, from 10.9% in 2013 to 12.4% in 2018 [23]. Investigating the reason for the delayed presentation of PDR is crucial for preventing DR-related blindness.

Therefore, we planned to study a group of PDR patients who had not received timely treatment by using a questionnaire survey to investigate the reasons for delaying DR diagnosis and treatment and evaluate the control of diabetes, treatment of DR, and severity of DR. Finally, we aimed to analyze the relevant factors and reasons for delaying diagnosis and treatment in these PDR patients.

## Material and methods

### Questionnaire design (Appendix 1)

In this study, a questionnaire survey method was used to investigate patients. The questionnaire gathered information about the patient's medical history, medication usage for systemic diseases, and ocular treatment they received. The questionnaire then focused on understanding the reasons for delayed PDR presentation among the patients and was divided into four distinct sections:

a. Knowledge about DR section: This part assessed the patients' understanding of DR-related knowledge before their PDR diagnosis. The assessment utilized a yes/no question format to evaluate the patients' knowledge.

b. Attitude toward DR treatment section: This section aimed to gain insights into the patients' attitudes toward the diagnosis and treatment of DR. It also aimed to identify subjective reasons for treatment delay by providing a list of various reasons for patients to choose from. Additionally, patients were asked whether they had ever refused DR treatment.

c. Difficulties in real-life section: This section investigated the social, economic, and systemic factors that patients believed could impact their willingness to seek DR diagnosis and treatment. It included questions related to insurance coverage, financial income situation, level of family support, and whether the patients were influenced by factors such as work, caregiving responsibilities, or other systemic diseases. Multiple-choice selection was utilized in this section.

d. Medical processes section: This section explored the patients' experiences with different medical processes, explicitly focusing on interactions with internal medicine physicians and ophthalmologists. It included inquiries about whether the internal medicine physicians or ophthalmologists provided DR education and informed their patients about the importance of regular fundus examinations or check-ups.

By structuring the questionnaire into these four sections, the study aimed to comprehensively investigate patients' knowledge, attitudes, obstacles, and experiences with medical processes related to DR diagnosis and treatment.

## Cross-sectional survey

The population to be surveyed in this study consisted of consecutive patients diagnosed with PDR during outpatient visits to our hospital between January 2021 and December 2021 who did not receive timely treatment. The study followed the tenets of the Declaration of Helsinki and the protocol and was approved by the Institutional Review Board of Beijing Tongren Hospital. Patients were required to sign an informed consent form, and written informed consent was obtained from each participant before commencing the survey.

Inclusion criteria: 1) patients had to be between 18 and 70 years of age; 2) patients must have been diagnosed with PDR requiring photocoagulation or vitrectomy in at least one eye; 3) patients had to show a delayed PDR presentation, defined as those who were diagnosed with PDR but did not receive treatment during their initial outpatient visit or those who received treatment but did not undergo timely follow-up, failing to detect disease progression; 4) patients must have completed all laboratory tests (including routine blood, urine, and renal function tests) or have had a precise diagnosis of systemic complications; 5) patients must have completed the questionnaire survey; 6) patients must have agreed to participate in the study.

Exclusion criteria: 1) patients with severe vitreous hemorrhage (VH) for which it could not be determined if PDR was the cause; 2) patients in whom both eyes had undergone vitrectomy due to PDR, making it impossible to determine if PDR was delayed before the surgery; 3) patients who currently did not require photocoagulation or vitrectomy but only needed close follow-up; 4) patients who were unable to complete the survey questionnaire; 5) patients with comorbidities such as glaucoma, optic nerve disease, or retinal detachment, as these conditions may cause irreversible vision loss; 6) patients who were in lockdown during the COVID-19 pandemic in Beijing and were excluded from the study due to the limited availability of appointments for DR patients, which could have resulted in delays in their treatment; 7) pregnant patient.

The selected patients underwent a comprehensive ophthalmic examination, including a corrected visual acuity (VA) test, intraocular pressure test, and slit-lamp assessment to evaluate the presence of neovascularization of the iris (NVI) and indirect ophthalmoscopy and vitrectomy if needed to determine the severity of PDR. PDR patients were diagnosed as those who required only photocoagulation or required vitrectomy. In addition, the presence of combined NVI or NVG was recorded.

The patients then completed the questionnaire survey. The general condition, overall health status, and history of ocular treatment details were recorded as follows: the diagnosis of combined hypertension; combined diabetic complications; history of participating in a DR screening program; the time between the last screening and diagnosis if they participated; the refusal of DR screening or PDR treatment recommendations; history of phacoemulsification cataract extraction, intravitreal injection (IV) of anti-vascular endothelial growth factor (VEGF) agents, or pan-retinal photocoagulation (PRP) treatment; and whether the patients were followed up on time after examinations or treatments. The results of the four-section questionnaire on reasons for delayed presentation of PDR were recorded.

The laboratory examination on HbA1C and questionnaire items on controlling blood glucose were recorded for further analysis of glucose control status. The renal function test result was recorded to determine the presence of chronic kidney disease (CKD) with impaired renal function.

Incomplete laser treatment was defined as adding 500 or more laser spots during outpatient photocoagulation or surgical procedures.

PDR patients with neovascularization elsewhere (NVE) but without high-risk PDR were only treated with PRP.

PDR that required vitrectomy was defined by the presence of any of the following conditions: 1) VH that hinders further examination of the fundus, confirmed as PDR being the cause of VH during surgery, 2) fibrovascular membranes involving or threatening the macula or associated with recurrent vitreous hemorrhage, and 3) VH due to PDR combined with NVI or NVG.

In cases where treatment was required for both eyes, the eye with better visual acuity and a less severe PDR was selected as the study eye.

## Grouping

The patient's acceptance of their DR diagnosis and treatment is directly related to their awareness of having DM. Therefore, there are inevitably differences in the reasons for delayed presentation, DR-related knowledge, and DM control between PDR patients with undiagnosed DM and those with a clear history of DM who did not receive timely PDR diagnosis and treatment. The former group has never received diabetes treatment before being diagnosed with PDR, while the latter often faces issues causing delays in PDR diagnosis and treatment during the DM treatment process despite being diagnosed with diabetes early on. In the subsequent investigation of the population known to have DM, we further classified them based on the following question in the survey questionnaire: "Have you ever refused the doctor's recommendation for DR diagnosis and treatment?" This latter group was divided into a non-compliant group (refusing DR diagnosis and treatment) and a compliant group (accepting DR screening or therapy). The compliant group included patients who underwent screening but were misdiagnosed as not having PDR and patients who received photocoagulation treatment but still had worsening conditions.

1. PDR patients with undiagnosed DM: Defined as patients who were unaware of having DM during initial ophthalmic consultation and their retinal abnormalities discovered were leading to a diagnosis of DR

2. PDR patients with a clear history of DM at the time of PDR diagnosis: This group of patients was further divided into two subgroups:

2.1. Patients who were able to complete DR screening and treatment as required: Defined as patients who have previously undergone DR screening and treatment as instructed by their doctor before the current visit, including one of the three following conditions:

2.1.1. Patients with underdiagnosed PDR: Defined as patients with a confirmed history of DM who underwent DR screening within the last six months as required but had negative findings at the latest visit and were diagnosed with PDR during the current visit.

2.1.2. Patients with PDR progression despite previous PRP: Defined as patients with a confirmed history of DM and DR who had previously undergone PRP as required but whose condition progressed to the extent that extra photocoagulation or surgical treatment was needed.

2.2. Patients who refused to participate in DR screening or treatment: Defined as diabetes patients previously advised to undergo DR screening or treatment but refused.

## Statistical analysis

Statistical analysis was performed using R version 3.20 (http://www.R-project.org). Patient characteristics were retrieved from their medical charts and recorded in Epidata Entry Client-version2.0.3.15 (http://epidata.dk). The mean and standard deviation (SD) were calculated for

continuous variables with a normal distribution. The median with quartiles was calculated for continuous variables with nonnormal distribution. The t-test or Mann–Whitney U test was carried out for continuous variables. The chi-square or Fisher's exact test was carried out for discrete data.

To investigate the factors related to patients refusing to undergo DR screening or treatment, two independent-sample comparisons were carried out on characteristics between patients who could complete DR screening and treatment as required and patients who refused to participate in DR screening and treatment. Variables with a p-value less than 0.3 were further enrolled in a binary backward stepwise logistic regression model. Each time, one variable was included or excluded from the model by comparing the Akaike information criterion (AIC) value; the model with the lowest AIC was chosen.

## Results

### General characteristics

A total of 157 patients, with a mean age of 50.0 ± 11.6 years and including 84 male patients (53.5%), were included in this study. The overall glycemic control was poor, with a median HbA1c of 7.8% (IQR 2.5%).

Most eyes (144, 91.7%) required vitrectomy intervention; 17 developed NVG, while only 13 required additional PRP.

PDR was confirmed in most contralateral eyes (154, 98.1%), with 3 cases of blindness due to trauma. Most of these eyes did not receive timely and effective treatment, resulting in 18 cases of blindness due to PDR, five patients treated with vitrectomy, 42 cases treated with PRP, and 10 cases treated with IV anti-VEGF agents (details in Table 1).

### Patient-related reasons for delayed presentation

**Patients with undiagnosed DM at the diagnosis of PDR.**   We identified 36 patients who were unaware of their DM diagnosis at the time of PDR diagnosis. The primary reason for delayed DR presentation was that these patients were unaware they had DM (36, 100.0%). Additionally, some patients believed that ophthalmic examination was only necessary when VA significantly declined or had a noticeable impact on their daily lives (14 cases, 38.9%). Furthermore, a few patients believed traditional Chinese herbal therapy could cure DM and did not undergo systematic evaluation even after receiving suggestions to rule out DM (4 cases, 11.2%). Notably, 14 (38.9%) did not have medical insurance, seven (19.4%) had economic difficulties, six (16.7%) had caregiving responsibilities, five (13.9%) had trouble taking time off from work, and 5 (13.9%) had concurrent management of other systemic diseases. These factors were identified as contributing to delayed medical attention. None of the patients had undergone a physical examination in the past year. Due to patients' lack of awareness of their DM diagnosis, we did not investigate their compliance with blood glucose control and knowledge of DR.

**Patients with a clear history of DM at the diagnosis of PDR.**   We surveyed 121 PDR patients with a known history of DM to explore the reasons and personal factors contributing to their delayed DR treatment. The survey included patients' acknowledgment of DR, compliance with DM management, and attitudes toward DR diagnosis and treatment. Some patients were not opposed to DR treatment but had other issues, while others directly resisted DR treatment. We conducted separate investigations for these two groups.

First, we identified common issues among patients with a known history of DM who experienced delays in PDR diagnosis and treatment:

**Table 1. Characteristics of diabetes patients in different groups.**

| | Patients with undiagnosed DM (36) | Patients with a history of DM (121) | | |
| --- | --- | --- | --- | --- |
| | | Patients who refused to participate in DR screening or treatment (75) | Patients who were able to complete DR screening and treatment as required | |
| | | | underdiagnosed PDR (30) | Progressed after PRP (16) |
| age (y, mean±SD) | 46.3±11.0 | 51.6±11.0 | 47.5±11.8 | 54.9±13.3 |
| DM duration (y, median, IQR) | 0.6, 1.1 | 13.0, 6.5 | 15,7.8 | 20, 4.5 |
| HbA1c (%, IQR) | 7.2, 2.5 | 7.6, 2.6 | 8.6,2.7 | 7.5, 2.1 |
| TCM without other medication (n) | 2 | 2 | 1 | 1 |
| HTN (n) | 25 | 35 | 18 | 9 |
| CKD (n) | 11 | 17 | 20 | 5 |
| coronary heart disease (n) | 9 | 10 | 9 | 4 |
| stroke (n) | 3 | 6 | 5 | 3 |
| Ocular symptom (m, median, IQR) | 3.5, 9.3 | 3.0, 7.0 | 4.0, 4.0 | 4.5, 10 |
| Cataract extraction without DR screening (n) | 6 | 1 | 1 | 0 |
| IV-anti VEGF (n) | 3 | 4 | 3 | 2 |
| incomplete PRP (n) | 6 | 7 | 5 | 7 |
| complete PRP (n) | 2 | 4 | 2 | 3 |
| Further treatment | | | | |
| required PRP (n) | 3 | 6 | 1 | 3 |
| required vitrectomy without NVG (n) | 30 | 42 | 27 | 12 |
| required vitrectomy with NVG (n) | 3 | 7 | 2 | 1 |

SD: Standard deviation, IQR: Interquartile range, HbA1c: Glycated form of hemoglobin, HTN: Hypertension, CKD: Chronic kidney disease, IV: Intravitreal injection, VEGF: Vascular endothelial growth factor, PRP: Pan-retinal photocoagulation, NVG: Neovascular glaucoma, TCM: Traditional Chinese medicine.

In terms of DR awareness, the patients generally exhibited inadequate knowledge. Only 29 (24.0%) individuals knew that diabetes patients require regular DR screening, and only 33 (27.3%) knew that untreated DR could lead to blindness.

In terms of behavior, patients attributed the primary reason for the delay to their excellent VA, leading them to believe that DR screening was unnecessary. Ninety-eight (81.0%) patients thought their good VA was the main reason for delaying DR diagnosis and treatment.

Regarding compliance with DM management, most patients (113/121, 93.3%) reported regular use of diabetic medication, but their glycemic control was poor. Among them, 22 patients (18.2%) had an HbA1c level above 6.5% at their visit.

Furthermore, we revealed the characteristics of patients who refused DR screening or treatment and the reasons for the refusal.

The two groups of patients showed similar levels of DR awareness in the survey regarding reasons for delay (p = 0.41, p = 0.83), and there were no significant differences in social support (p = 0.98) or insurance coverage (p = 0.99).

Compared to diabetes patients who did not refuse the recommended DR screening and treatment, those who refused exhibited the following characteristics: a lower proportion of regular visits to internal medicine clinics (43/75, 42/46, p<0.001), a shorter duration of diabetes (13, 16, p = 0.005), a higher proportion of patients with negative findings during DR screening

but without regular follow-up recommendation (49/75, 14/46, p<0.001), and a lower proportion of patients who can tolerate discomfort during DR examination or treatment (17/75, 3/46, p = 0.02). Furthermore, logistic regression analysis revealed three factors associated with the refusal of DR diagnosis and treatment:

A.  Inability to receive regular diabetes treatment in internal medicine clinics (OR 6.78, 95% CI 1.73–26.59, p = 0.006),

B.  Inability to tolerate the discomfort during ophthalmic examination and treatment (OR 15.15, 95% CI 2.70–83.33, p<0.001),

C.  Patients who did not detect any retinal abnormalitieswith negative findings in DR screening and who were not informed about the need for regular screening (OR 2.05, 95% CI 1.36–3.09, p<0.001) (AIC = 118.88, AUC = 0.861).

### Patient-perceived reasons for delayed presentation related to medical, family, and social aspects

After investigating the reasons for the delay as perceived by themselves, we further examined the difficulties that could lead to delays, including financial problems, work distractions, family caregiving responsibilities, inability to seek medical attention independently, and the need to treat other systemic diseases. The ranking of these reasons varied among different patient groups (Fig 1).

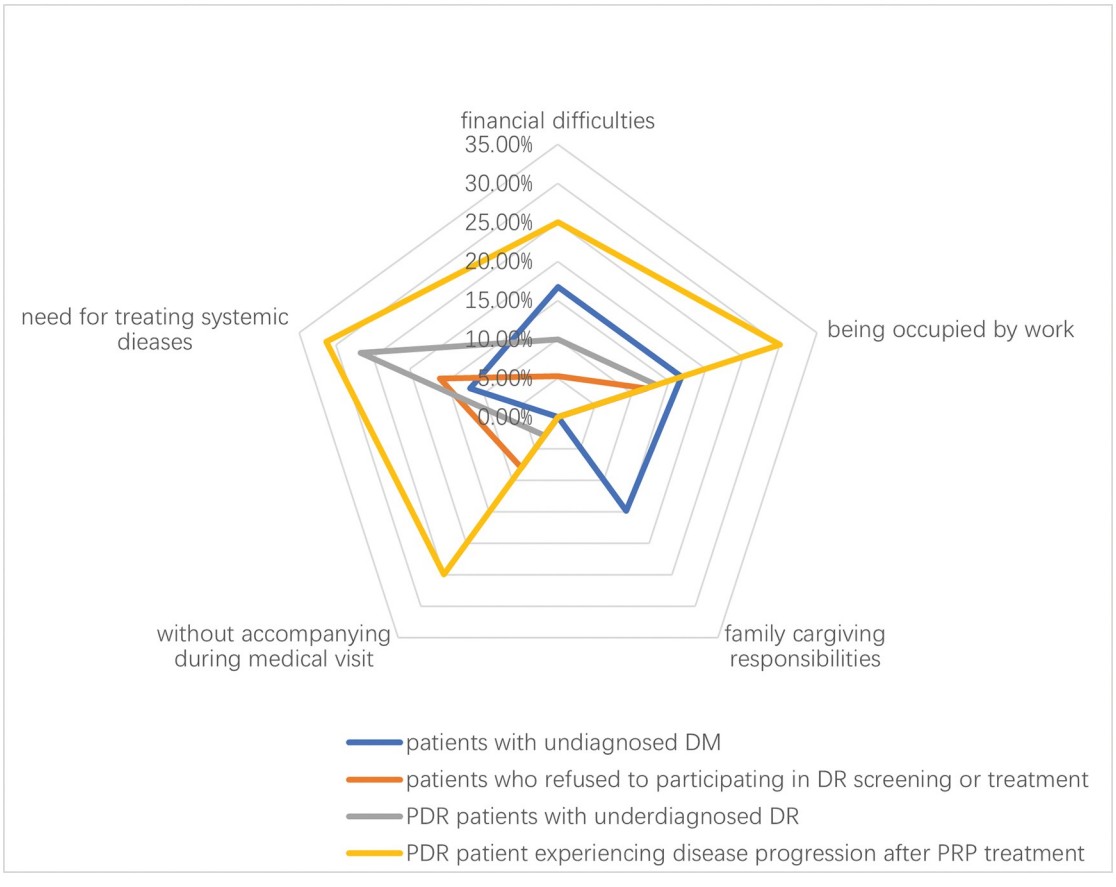

**Fig 1. The radar chat for the patient-perceived reasons for delayed presentation among different groups.**

1. For individuals who were unaware of their DM diagnosis at the time of PDR diagnosis, the factors influencing their DR treatment were as follows: financial difficulties (16.7%), being occupied by work (16.7%), family caregiving responsibilities (14.9%), and the need for treatment of other systemic diseases (11.9%).

2. For the PDR patients who refused DR treatment suggestion, the factors influencing their participation in DR treatment were as follows: the need for treatment of other systemic diseases (16.0%), being occupied by work (12.0%), lack of accompanying during medical visits (8.0%), and financial difficulties (5.3%).

3. For the PDR patients with underdiagnosed DR, the factors influencing their situation were as follows: the need for treatment of other systemic diseases (26.7%), being occupied by work (13.3%), financial difficulties (10.0%), and lack of accompanying during medical visits (2.9%).

4. For the PDR patient experiencing disease progression after PRP, the factors influencing their situation were as follows: the need for treatment of other systemic diseases (31.3%), lack of accompanying during medical visits (25%), financial difficulties (25%) and being occupied by work (30%).

## Deficiencies in the healthcare services in response to the delayed presentation

In addition to the patients and their support system, we identified deficiencies in DM screening, the management of diabetes patients by internal medicine physicians, and the treatment for DR provided by ophthalmologists, all contributing to delays in DR management.

## Inadequate DM screening among the general population

Our study identified 36 patients diagnosed with DM only when PDR was detected in the ophthalmology clinic. These patients had not undergone DM screening previously. For them, the main issue contributing to treatment delay was why DM was not detected promptly.

## Issues with internal medicine physicians in DR management

After being diagnosed with DM, patients most frequently interact with internal medicine physicians. Among patients with a known history of DM but who experienced treatment delays, we identified problems related to guidance on regular visits, education on DR, and the sensitivity of DR screening by internal medicine physicians.

Internal medicine physicians did not emphasize the importance of regular follow-up visits for DM patients. Our study found that among individuals with a known history of DM, 36 patients (29.75%) did not have regular visits to internal medicine physicians. Furthermore, the proportion of patients who did not have regular visits and underwent DR screening was lower. This was evident in the following aspects: 1) a lower proportion of patients who received information from internal medicine physicians about the potential blindness caused by DR, compared to those with regular visits (4/36, 29/85, p = 0.01); 2) a lower proportion of patients with irregular visits who underwent DR screening, compared to those with regular visits (11/36, 67/85, p = 0.01).

We found a lack of DR education among internal medicine physicians. Among the 121 patients known to have DM, only 18 individuals (14.9%) received DR education from internal medicine physicians, which was significantly lower than the proportion of patients receiving DR education from ophthalmologists (18/121 vs 62/121, p<0.001). Even among the 85 patients with regular visits, only 25 individuals were aware of the importance of regular fundus

examinations. The majority of patients obtained knowledge about DR through ophthalmic DR screening, and those who underwent DR screening had better awareness of DR compared to those who did not undergo screening (DR screening necessary: 40/66 vs 81/91, p<0.001; DR can cause blindness: 46/66 vs 80/91, p = 0.008).

During the initial visit for DM, internal medicine physicians did not sufficiently emphasize the potential for blindness due to DR and the importance of regular asymptomatic DR examinations. Among the 55 patients who refused DR screening, 87.3% believed that good vision meant they did not need screening, which was the primary reason for their refusal. Additionally, we found that 14.5% of patients delayed DR screening out of fear of detecting retinal abnormalities.

We observed a low sensitivity of DR screening initiated by internal medicine physicians. This could lead patients to mistakenly believe that their fundus were normal and overlook the importance of regular follow-up examinations. Our study identified 30 patients who underwent DR screening within the past six months without detecting any retinal abnormalities. However, within the subsequent six months, 29 patients developed PDR retinal lesions requiring surgical intervention, including two individuals who developed NVG. Among them, 20 patients had undergone DR screening initiated by internal medicine physicians, while 10 had undergone screening initiated by ophthalmologists. The rate of missed diagnosis in DR screening initiated by internal medicine physicians was 20/21, significantly higher than that of ophthalmologists, which was 10/45 (p<0.001).

Internal medicine physicians did not adequately emphasize the need for regular DR screening when DR was not detected. Our findings demonstrated that among patients who underwent initial screening without detecting DR, those who were not informed about the importance of regular follow-up had fewer opportunities for PDR treatment compared to those who were notified. Among the 30 patients with negative retinal findings during the initial screening, 14 individuals were not informed about the need for follow-up, and 12 were not informed by internal medicine physicians. These patients did not receive PDR treatment during the subsequent six months. On the other hand, 16 patients were informed about the need for regular follow-up. During the subsequent examinations, they were diagnosed, with two individuals receiving the opportunity for complete PRP and five individuals completing partial PRP before the onset of vitreous hemorrhage (0/14, 7/16, p = 0.007).

Our investigation revealed that 28 patients experienced delays in DR treatment due to hospitalization for systemic diseases. These patients did not receive ophthalmic consultations or treatment during their hospital stay, nor did they receive any educational information about DR from internal medicine physicians.

We identified 8 PDR patients who regularly sought treatment from traditional Chinese medicine (TCM) practitioners. These patients were of working age, ranging from 33 to 62 years old, with a median age of 59 (standard deviation 7.75 years). However, these patients relied solely on TCM instead of receiving systematic glucose-controlling medicine (6/8), resulting in poor blood glucose control (elevated HbA1c, with an average of 7.1±1.1) and severe systemic complications (8/8). Prior to their diagnosis of PDR, these patients lacked awareness of DR, unaware of its potential to cause blindness and the need for regular examinations (8/8). Furthermore, none underwent DR screening (8/8), leading to the development of PDR complications requiring vitrectomy.

**Insufficiencies in the diagnosis and Treatment of DR by Ophthalmologists.** Ophthalmologists are responsible for treating DR, particularly in emphasizing the importance of regular follow-up for early-stage DR that does not affect vision. However, we found that the DR patient education by ophthalmologists was inadequate. Only 62 out of 121 patients received

DR education from ophthalmologists. Additionally, the false thought that having good VA meant they did not require ophthalmic care and postponed seeking medical attention was commonly presented in patients who failed to receive DR treatment and those who complied with the recommended DR treatment (59/75, 39/46, p = 0.55).

Ophthalmologists have shown insufficient efforts in educating and encouraging patients who exhibit resistance and pain-related concerns during DR treatment. Alternatively, understanding DR treatment and follow-up among diabetes patients may have needed to be improved. We discovered that discomfort experienced during fundus examinations and PRP became the primary reasons for delayed presentation of DR. When investigating the reasons behind delayed PRP in patients with DR, we found that 13 (65%) patients who refused PDR treatment did so out of fear of pain associated with PRP. Furthermore, four individuals (20%) refused PRP due to discomfort of fundus examination. Similarly, among 16 patients who experienced disease progression after PRP, four (25.0%) declined and refused to complete the treatment due to pain related to PRP. Three patients (18.8%) declined subsequent follow-ups due to discomfort during eye examinations post-PRP. 50% of those who refused DR treatment and 66.7% of those who failed to stick to the standard follow-up plan after PRP exhibited insufficient awareness regarding the risk of blindness associated with DR and the necessity of regular follow-up examinations. As a result, they lacked the necessary determination to overcome discomfort and complete the required examinations and treatment.

We found that ophthalmologists needed to adequately emphasize the possibility of disease progression and the importance of regular follow-up examinations after PRP, as demonstrated by the patient's lack of knowledge. Among the 16 patients who completed PRP, 11 were unaware of the need for regular follow-up examinations after treatment. Despite relatively stable vision in these patients (11/16), factors such as being occupied by work, systemic illnesses, family responsibilities, financial difficulties, and discomfort with examinations and PRP contributed to their failure to attend follow-up appointments. As a result, ten patients experienced disease progression to a stage requiring vitrectomy.

Furthermore, we found that some ophthalmologists performed PRP with inadequate coverage, leading to delays in disease management. Six patients were informed of completing PRP, and their condition progressed to PDR requiring vitrectomy. However, during the vitrectomy, it was discovered that the PRP coverage was insufficient. These patients did not attend follow-up examinations promptly after completing PRP.

Some ophthalmologists overlooked DR screening when performing cataract extraction surgery on diabetes patients. Among the ten patients who underwent cataract extraction surgery before receiving DR treatment, seven individuals with improved vision underwent no DR screening. In comparison, only one out of the three patients whose vision did not improve underwent DR screening, leading to a missed diagnosis. The improvement in VA following cataract extraction surgery may have contributed to the delay in the diagnosis and treatment of DR.

## Discussion

Early detection and intervention can prevent blindness caused by DR [4,5], however delayed treatment may result in severe and irreversible consequences. Emerging evidence suggests that chronic inflammation may play a role in DR progression and possibly make the untreated PDR worse in patients failed to be identified in time [10–14]. While established methods for detecting DM [1] and DR [2,3] exist, this research sheds light on the complex and diverse factors contributing to delayed presentation of DR in Chinese patients, an area previously unexplored.

## The specific DM population that the ophthalmologist should pay more attention to

Our study identified several specific patient populations requiring closer attention from ophthalmologists.

1. Patients with poor diabetes control:

Our investigation revealed that this group of diabetes patients commonly exhibited poor blood glucose control with an average HbA1c of 7.8% and only 18.2% achieving the recommended HbA1c level of below 6.5%', which may lead to rapid DR progression. Previous studies support this association [10,11,24], highlighting the increased risk of DR development in individuals with elevated HbA1c levels [13,25]. Therefore, ophthalmologists should prioritize education and intervention for this specific patient population.

2. Patients with a high possibility of refusing DR treatment:

A. Our study analyzed factors associated with treatment refusal. Ophthalmologists should be attentive to patients with the following characteristics, as they are more likely to be less engaged in DR treatment. Targeted education and follow-up should be provided to these patients to slow the progression of DR. Patient information leaflets or website education may also allow patients to review and revise the issues.Patients who lack regular internal medicine visits:. Our research revealed a 6.78 times higher chance of receiving timely DR treatment for patients with regular internal medicine visits compared to those without. This could be attributed to regular patients receiving more DR education (29/85, 4/36) and more DR screening opportunities (67/85, 11/36).

B. Patients intolerant of pain and discomfort: Compared to individuals who were less resistant, those who strongly resisted pain and discomfort were 15.15 times more likely to refuse DR treatment. This resistance is particularly evident among patients who refuse PRP (13/20), whereas factors related to delays in PRP due to hospitalization for systemic diseases accounted for only 5/20 cases. Surveys regarding satisfaction with PRP indicate that insufficient knowledge about laser treatment, discomfort caused by pupillary dilation, pain during laser treatment, and failure to achieve expected visual acuity may lead to patient reluctance towards PRP, thereby affecting compliance with PRP and subsequent follow-up examinations. The results of our study align with previous research.

C. Patients unaware of the need for regular follow-up after DR screening and with no positive DR findings: These patients were 2.05 times more likely to refuse DR screening or treatment compared to those undergoing regular follow-up with DR detection. Guidelines recommend annual retinal examinations for diabetes patients without DR [3]. In our study, a group of patients (30/121) had no DR detected during their initial screening but progressed to PDR within the following six months. Most of these patients' DR screenings were conducted by internal medicine doctors (20/30), indicating the possibility of missed diagnoses. However, these patients shared the common characteristics of a long time of poor blood glucose control, with a median HbA1c of 8.6 and a median duration of DM of 15 years. It was suggested that individuals with a long duration of DM and poor blood glucose control might experience rapid progression of DR [13], even if abnormalities are not initially detected. Annual follow-up examinations may delay the diagnosis, indicating the need to shorten the interval for DR screenings. In addition, emerging indicators have been identified to be related to the development DR

[26,27], the patients without positive DR screen findings may take the idicators test for evidence of more closely follow-up.

**Reducing delays requires joint efforts from society and healthcare professionals.** We identified social factors influencing timely DR treatment access, including financial constraints [16], lack of nearby facilities, competing commitments and and hospitalization [15,20]. Addressing these barriers requires collaborative efforts from healthcare providers, policy-makers, and community organizations.

Low-income individuals without health insurance often lack access to diabetes screening, potentially delaying diagnosis and DR detection. This is particularly concerning for younger patients, where 38.9% lacked insurance and 19.4% cited financial barriers to medical care. Many patients missed medical visits due to obligations such as caring for family members or work, and these patients had not undergone regular health check-ups in the past year. Initiatives should focus on enhancing diabetes screening for uninsured and low-income populations to identify individuals with diabetes early and facilitate subsequent DR screenings.

The absence of a companion for medical visits may pose a barrier for diabetes patients to attend DR treatment. This situation is particularly prominent among patients who refuse DR treatment (6/20) and those who have completed PRP but have no regular follow-up examinations (4/16). As PDR can impact patients' independent living abilities, causing inconvenience in their daily lives, encouraging family members to accompany patients with visual impairments can improve treatment access and compliance.

The inability to take time off from work may hinder diabetes patients from attending DR treatment. Our research identified the inability to take time off work as a barrier to DR screening (10.9%) and follow-ups (13% missed diagnosis, 18.7% post-PRP). During initial consultations, identifying concerns about work leave is crucial. Individualized guidance and support, including physician education on the impact of end-stage DM on work attendance, can enhance treatment compliance.

**Medical care.**

a. Our analysis revealed that delayed treatment for diabetic retinopathy (DR) can be attributed to both patient- and physician-related factors. Effective DR management requires a coordinated and seamless chain of care, encompassing initial DM diagnosis, follow-up visits, DR screening, and ultimately, diagnosis and treatment of DR. Disruptions at any point in this chain can lead to treatment delays and potentially detrimental consequences for patients' vision. Therefore, it is paramount for physicians and ophthalmologists to vigilantly monitor and manage every stage of this care pathway, ensuring timely and comprehensive care for individuals with diabetes.Both physicians and ophthalmologists need to improve their diabetes patient education, particularly regarding the importance of DR screening regardless of vision. Our research revealed that 81.0% of delayed patients believed that good vision meant they did not need medical attention. This misconception was prevalent among patients receiving regular internal medicine care (60/85) and those undergoing ophthalmic DR treatment (59/75). Studies have shown that patients' understanding of the potential blindness caused by the delayed detection of DR directly influences their adherence to DR treatment. Insufficient awareness of the need for DR screening and the misconception that good vision eliminates the need for ocular examinations are associated with the severity of PDR when diabetes patients seek ophthalmic care. Therefore, patient education for diabetes patients should focus on the importance of regular follow-ups for early DR detection, irrespective of visual symptoms. During consultations with diabetes patients, every opportunity

should be seized to emphasize that early-stage DR may not affect vision, the need for regular DR screening, and the significance of early detection and treatment.

b.  Some errors made by the physicians:

Internal medicine physicians should prioritize DR diagnosis and treatment alongside managing systemic conditions. This study's findings align with previous research, highlighting the association between hospitalization and delayed DR treatment [20]. Therefore, timely ophthalmological consultations during internal medicine care are crucial for early DR detection and intervention.

During DM follow-ups, internal medicine physicians should pay particular attention to patients experiencing anxiety about DR diagnosis. This study identified a previously unreported factor, where 14.5% of patients refused screening due to fear of positive results. Even with some understanding of DR, these patients avoid screenings due to excessive anxiety. Notably, depression and anxiety rates are high among Chinese DR patients (25% and 13.5%, respectively), and social support can effectively alleviate these issues [28]. To address this, internal medicine physicians should promptly identify patients with psychological concerns and provide positive guidance. This includes informing them that DR does not lead to inevitable blindness and that early treatment can effectively manage the condition. Additionally, seeking cooperation from family members and encouraging patient participation in DR diagnosis and treatment are essential.

Physician-initiated DR screenings require improvement, with our study showing a concerning 20/21 missed diagnosis rate. While previous research suggests a potential role for physicians in DR detection with a sensitivity of 86.6% [29], our findings highlight a significant gap. Recent studies demonstrate that incorporating fundus photography and remote image reading with physician-initiated screenings can significantly enhance accuracy [27,30]. Given that most diabetic patients first encounter physicians, equipping them with improved screening techniques and remote resources even artificial intelligence aliened result reading is critical for timely DR detection and intervention.

Among the 20 PDR patients with missed diagnoses, 14 had good vision but lacked awareness of the need for regular follow-up visits. This highlights the need for physicians and other healthcare professionals to strengthen patient education and emphasize regular ophthalmic care, regardless of vision status.

Additionally, we identified eight patients receiving traditional Chinese medicine (TCM) who opted for it instead of conventional diabetes medication. This resulted in poor blood sugar control, no DR screenings, and delayed diagnosis until severe PDR developed. These patients, aged 32–62 and belonging to the working population, also suffered from uncontrolled systemic complications. Previous studies have shown that traditional Chinese medicine (TCM), can not take place of the conventional treatment for DM [31,32]. Their case underscores the importance of improvingTCM physicians' understanding of DR screening and educating patients about the need for medication, diet, exercise, and regular check-ups, even in the absence of vision changes.

c.  The ophthalmologists should make some improvements:

Ophthalmologists' failure to remind patients who have undergone cataract extraction about the importance of DR screening can delay treatment for diabetes patients who experience improved vision post-cataract extraction. Ten patients with a history of cataract extraction were included in this study, of which eight did not undergo DR screening. Among these eight patients, seven reported improved vision after the surgery, and five of them did not receive a fundus examination because they believed their vision was already good. Previous research has

indicated that vision improvement following cataract surgery in diabetes patients can potentially mask vision symptoms caused by DR. Additionally, cataract surgery in diabetes patients can contribute to the progression of PDR, necessitating perioperative intravitreal anti-VEGF agents and vigilant postoperative follow-up [33,34]. Therefore, ophthalmologists should prioritize strengthening DR screening and follow-up for diabetes patients undergoing cataract surgery. This approach is crucial to identify DR promptly and prevent further deterioration of vision due to DR.

Ophthalmologists should pay particular attention to patients who cannot tolerate the discomfort of fundus examination or PPP treatments. Since the patients may fail to attach to the treatment or follow-up plan, regular DR screening and standard complete PRP should be emphasized, and communication with the patient companions to encourage the patients for further DR treatment should be strengthened. Retrobulbar anesthesia may be applied to patients who cannot tolerate the pain of PRP.

Ophthalmologists must prioritize regular follow-ups and effectively communicate to patients undergoing PRP about ongoing examinations' importance. Our study observed that 16 patients with PDR had completed PRP, but their condition continued progressing. Among these patients, 11 were unaware of the necessity for regular post-PRP examinations, and ten believed their vision had stabilized, leading them to forego further treatment. Unfortunately, this delay in disease detection hindered early intervention. Furthermore, we discovered that six patients needed complete PRP and were not frequently required to attend follow-up examinations. Consequently, their condition progressed, ultimately requiring vitrectomy surgery. Previous research has demonstrated that effective PRP can prevent 50% of PDR progression and mitigate 50% of severe vision damage caused by PDR [35]. However, despite PRP, 42% of PDR patients still experience disease progression [36]. Patients who initiate PRP after experiencing vitreous hemorrhage and retinal detachment are more likely to encounter DR progression following PRP [36,37]. Therefore, for diabetes patients who have completed PRP, ophthalmologists should strengthen education regarding the effectiveness of PRP and stress the need for regular follow-up examinations. It is crucial to adhere to standardized PRP protocols meticulously.

## Limitations

This study is a cross-sectional survey that focused on PDR patients who delayed seeking medical treatment. The study specifically excluded patients with well-controlled glucose levels and those compliant with DR treatment, and we could not investigate the delayed presentation in those patients. Notably, this study's proportion of patients receiving intravitreal anti-VEGF agents was relatively low. Furthermore, most patients who received this treatment did so after experiencing vitreous hemorrhage or NVG. Therefore, the reasons for the delayed progression of PDR following intravitreal anti-VEGF agents remain unclear, indicating a need for further research. Additionally, it is worth mentioning that the number of patients who underwent PRP received intravitreal anti-VEGF agents or cataract surgery was relatively small. This may have resulted in a selection bias among the study population. Furthermore, regarding retinopathy assessment, this study did not include the measurement of VA. Instead, the patient's ocular condition was evaluated solely through fundus examination. Consequently, the study failed to demonstrate any VA impairment within this group of patients. Regarding systemic examination, the study exclusively recorded the patient's medical history of CAD, cerebral infarction, and other diseases and the results of kidney function tests. However, the urine albumin-to-creatinine ratio (ACR), an indicator for CKD, was not conducted, leading to a deviation in evaluating the systemic condition. The epicardial fat thickness [26] or neuregulin-4 [10],

serum C-reactive protein to albumin ratio [38], and indicators for the inflammation nature of DR [13] were not conducted, which may be involved in further studies investigating the factors related to the progression of DR after PRP. We did not involve gestational diabetes mellitus patients, we can not show the relationship between atrial conduction time and P-wave dispersion [39] and DR progression.

## Conclusions

This study investigated the factors contributing to delayed presentation among patients with PDR. A significant proportion of individuals in the delayed population were found to have undiagnosed DM. As a result, it is crucial to enhance DM screening efforts and promote education about the condition. Among patients already aware of their DM status, reasons for delay included insufficient knowledge about DR, negative attitudes towards screening and treatment, and difficulties seeking medical care in real-life situations. Furthermore, there needed to be more improvements in the detection, treatment, and follow-up of DR by both internal medicine practitioners and ophthalmologists. It is imperative to strengthen education regarding the importance of regular DR screening, especially when the VA has not severely deteriorated. Particular attention should be given to patients with irregular visits to internal medicine specialists and those concerned about discomfort during DR treatment. Additionally, patients who undergo cataract surgery, participate in DR screening conducted by physicians, utilize traditional Chinese medicine treatments, undergo complete PRP, and have negative screening findings should also receive regular follow-up examinations. Finally, vigilance to patients' challenges during the diagnosis and treatment is crucial. Patients should be encouraged to cooperate and receive support from family members throughout their treatment journey.

## Supporting information

**S1 File. The ethics committee approval by the Institutional Review Board of Beijing Tongren Hospital.** The Chinese original version and English translation of the ethics committee approval.
(DOCX)

**S2 File. The informed consent of this study.** The informed consent of this study in Chinese.
(DOCX)

**S3 File. The translation of the informed consent.** The English translation of the informed consent of this study.
(DOCX)

**S4 File. The translation of the questionnaire.** The English translation of the questionnaire of the study.
(DOCX)

## Acknowledgments

We would like to express our sincere gratitude to all those who contributed to the successful completion of this study. Their support, guidance, and encouragement have been invaluable throughout the research process.

First and foremost, we extend our heartfelt appreciation to the patients who participated in this study. Their willingness to share their experiences and insights was pivotal in shedding

light on the factors contributing to delayed presentation in patients with proliferative diabetic retinopathy.

We are grateful to our colleagues and fellow researchers who provided valuable feedback and suggestions during this study. Their expertise greatly enriched the quality of our research.

## Author Contributions

**Conceptualization:** Meng Zhao, Lin Zhang, Jun Xu, Jipeng Li.

**Data curation:** Meng Zhao.

**Formal analysis:** Meng Zhao, Jipeng Li.

**Investigation:** Meng Zhao, Lin Liu, Lin Zhang, Jun Xu, Jipeng Li.

**Methodology:** Meng Zhao, Jun Xu.

**Resources:** Lin Liu.

**Software:** Meng Zhao.

**Supervision:** Aman Chandra, Jipeng Li.

**Validation:** Meng Zhao, Jipeng Li.

**Visualization:** Meng Zhao.

**Writing – original draft:** Meng Zhao, Jipeng Li.

**Writing – review & editing:** Meng Zhao, Aman Chandra, Lin Liu, Lin Zhang, Jun Xu, Jipeng Li.

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
