## [Decision Letter · Decision Letter 0]

18 Sep 2023

PONE-D-23-26728Investigation of the reasons for delayed presentation in proliferative diabetic retinopathy patientsPLOS ONE

Dear Dr. Li,

Thank you for submitting your manuscript to PLOS ONE. After careful consideration, we feel that it has merit but does not fully meet PLOS ONE’s publication criteria as it currently stands. Therefore, we invite you to submit a revised version of the manuscript that addresses the points raised during the review process.

We look forward to receiving your revised manuscript.

Kind regards,

Jiro Kogo

Academic Editor

PLOS ONE

Reviewers' comments:

Reviewer's Responses to Questions

**Comments to the Author**

1. Is the manuscript technically sound, and do the data support the conclusions?

Reviewer #1: Yes

Reviewer #2: Yes

Reviewer #3: Yes

2. Has the statistical analysis been performed appropriately and rigorously? 

Reviewer #1: Yes

Reviewer #2: Yes

Reviewer #3: Yes

3. Have the authors made all data underlying the findings in their manuscript fully available?

Reviewer #1: Yes

Reviewer #2: Yes

Reviewer #3: Yes

4. Is the manuscript presented in an intelligible fashion and written in standard English?

Reviewer #1: Yes

Reviewer #2: Yes

Reviewer #3: Yes

5. Review Comments to the Author

Reviewer #1: There must be some revisions apply to the manuscript. The logic of the study is not clear in introduction. Therefore, more data must be added as background. Type 2 DM (Swiss Med Wkly. 2019;149:w20139. doi: 10.4414/smw.2019.20139) and its complications (Eur J Clin Invest. 2020;50(3):e13206. doi: 10.1111/eci.13206) are associated with inflammation. Specifically, diabetic neuropathy (Preprints 2023, 2023060202. doi:10.20944/preprints202306.0202.v1), diabetic retinopathy (Front Immunol. 2020;11:583687. doi: 10.3389/fimmu.2020.583687. eCollection 2020) and diabetic nephropathy (Postgrad Med. 2023;135(5):519-523. doi: 10.1080/00325481.2023.2214058) are characterized with high burden of inflammation. Thus, early detection of diabetes and its complications could improve effective management of the disease and its complications.

Moreover, discussion must be improved by discussing similar works. Some examples are https://doi.org/10.1007/s13410-021-01040-5 and https://doi.org/10.1007/s13410-022-01136-6

Reviewer #2: Dear Autors

I had the privilege of reviewing your manuscript.

The early diagnosis of DR and Diabetes are paramount in maintaining the patients quality of life, and your paper address some hurdles those patients have to achieve that.

Reviewer #3: 1. What is the definition of ”Delayed presentation of PDR”?

2. In this study, the rate of missed diagnosis in DR screening initiated by internal medicine physicians was 20/21, higher than that of ophthalmologists. Necessity of DR screening by internists rather than ophthalmologists need to be discussed by the authors.

3. It is an important issue for ophthalmologists that one of the reasons for the delayed presentation of DR is the inability to tolerate the discomfort during DR treatment. The author states that Particular attention should be given to patients concerned about discomfort during DR treatment. The authors need to discuss how to specifically conduct examinations and treatment.

6. PLOS authors have the option to publish the peer review history of their article (what does this mean?). If published, this will include your full peer review and any attached files.

Reviewer #1: No

Reviewer #2: **Yes: **Rodrigo Brant

Reviewer #3: No

---

## [Author Response · Author response to Decision Letter 0]

21 Oct 2023

Dear editor and reviewers:

 Thank you for your consideration and suggestion. We have made several changes according to your suggestion. 

I have checked the manuscript met the PLOS ONE’s style requirement.

The data is stored at zhao, meng (2023), “questionnaire for PDR delayed presentation”, Mendeley Data, V1, doi: 10.17632/z956gdk4f7.1

I had update my ORCID ID, as 0009-0003-2141-5782.

 I have checked the Methods section in lines 141-145, as The study followed the tenets of the Declaration of Helsinki and the protocol and was approved by the Institutional Review Board of Beijing Tongren Hospital. Patients were required to sign an informed consent form, and written informed consent was obtained from each participant before commencing the survey.

Reviewers' comments:

Reviewer's Responses to Questions

Comments to the Author

1. Is the manuscript technically sound, and do the data support the conclusions?

Reviewer #1: Yes

Reviewer #2: Yes

Reviewer #3: Yes

2. Has the statistical analysis been performed appropriately and rigorously?

Reviewer #1: Yes

Reviewer #2: Yes

Reviewer #3: Yes

3. Have the authors made all data underlying the findings in their manuscript fully available?

Reviewer #1: Yes

Reviewer #2: Yes

Reviewer #3: Yes

4. Is the manuscript presented in an intelligible fashion and written in standard English?

Reviewer #1: Yes

Reviewer #2: Yes

Reviewer #3: Yes

5. Review Comments to the Author

Reviewer #1: There must be some revisions apply to the manuscript. The logic of the study is not clear in introduction. Therefore, more data must be added as background. Type 2 DM (Swiss Med Wkly. 2019;149:w20139. doi: 10.4414/smw.2019.20139) and its complications (Eur J Clin Invest. 2020;50(3):e13206. doi: 10.1111/eci.13206) are associated with inflammation. Specifically, diabetic neuropathy (Preprints 2023, 2023060202. doi:10.20944/preprints202306.0202.v1), diabetic retinopathy (Front Immunol. 2020;11:583687. doi: 10.3389/fimmu.2020.583687. eCollection 2020) and diabetic nephropathy (Postgrad Med. 2023;135(5):519-523. doi: 10.1080/00325481.2023.2214058) are characterized with high burden of inflammation. Thus, early detection of diabetes and its complications could improve effective management of the disease and its complications.

Moreover, discussion must be improved by discussing similar works. Some examples are https://doi.org/10.1007/s13410-021-01040-5 and https://doi.org/10.1007/s13410-022-01136-6

Changes have been made as you suggested. They had been added to lines 81-85 in introduction section, lines 499-500,769-772 in discussion section.

Reviewer #2: Dear Autors

I had the privilege of reviewing your manuscript.

The early diagnosis of DR and Diabetes are paramount in maintaining the patients quality of life, and your paper address some hurdles those patients have to achieve that.

Thank you for your encouragement.

Reviewer #3: 1. What is the definition of ”Delayed presentation of PDR”?

The delayed presentation of PDR was in line 148-151. It was defined as delayed PDR presentation, defined as those who were diagnosed with PDR but did not receive treatment during their initial outpatient visit or those who received treatment but did not undergo timely follow-up, failing to detect disease progression.

2. In this study, the rate of missed diagnosis in DR screening initiated by internal medicine physicians was 20/21, higher than that of ophthalmologists. Necessity of DR screening by internists rather than ophthalmologists need to be discussed by the authors.

Changes had been made in lines 669-671. The necessity of DR screening by internists is added. 

3. It is an important issue for ophthalmologists that one of the reasons for the delayed presentation of DR is the inability to tolerate the discomfort during DR treatment. The author states that Particular attention should be given to patients concerned about discomfort during DR treatment. The authors need to discuss how to specifically conduct examinations and treatment.

Changes had been made in lines723-728, added the education, communication and local anesthesia.

6. PLOS authors have the option to publish the peer review history of their article (what does this mean?). If published, this will include your full peer review and any attached files.

Do you want your identity to be public for this peer review? For information about this choice, including consent withdrawal, please see our Privacy Policy.

Reviewer #1: No

Reviewer #2: Yes: Rodrigo Brant

Reviewer #3: No

We have uploaded the figure files that have been revised by PACE.

Looking forward to hearing from you soon.

Best regards,

Jipeng Li

Beijing Tongren hospital

---

## [Decision Letter · Decision Letter 1]

19 Nov 2023

PONE-D-23-26728R1Investigation of the reasons for delayed presentation in proliferative diabetic retinopathy patientsPLOS ONE

Dear Dr. Li,

Thank you for submitting your manuscript to PLOS ONE. After careful consideration, we feel that it has merit but does not fully meet PLOS ONE’s publication criteria as it currently stands. Therefore, we invite you to submit a revised version of the manuscript that addresses the points raised during the review process.

We look forward to receiving your revised manuscript.

Kind regards,

Jiro Kogo

Academic Editor

PLOS ONE

Journal Requirements:

Reviewers' comments:

Reviewer's Responses to Questions

**Comments to the Author**

1. If the authors have adequately addressed your comments raised in a previous round of review and you feel that this manuscript is now acceptable for publication, you may indicate that here to bypass the “Comments to the Author” section, enter your conflict of interest statement in the “Confidential to Editor” section, and submit your "Accept" recommendation.

Reviewer #1: All comments have been addressed

Reviewer #3: All comments have been addressed

2. Is the manuscript technically sound, and do the data support the conclusions?

Reviewer #1: Yes

Reviewer #3: Yes

3. Has the statistical analysis been performed appropriately and rigorously? 

Reviewer #1: Yes

Reviewer #3: Yes

4. Have the authors made all data underlying the findings in their manuscript fully available?

Reviewer #1: Yes

Reviewer #3: Yes

5. Is the manuscript presented in an intelligible fashion and written in standard English?

Reviewer #1: Yes

Reviewer #3: Yes

6. Review Comments to the Author

Reviewer #1: There must be some revisions apply to the manuscript. The logic of the study is not clear in introduction. Therefore, more data must be added as background. Type 2 DM (Swiss Med Wkly. 2019;149:w20139. doi: 10.4414/smw.2019.20139) and its complications (Eur J Clin Invest. 2020;50(3):e13206. doi: 10.1111/eci.13206) are associated with inflammation. Specifically, diabetic neuropathy (Preprints 2023, 2023060202. doi:10.20944/preprints202306.0202.v1), diabetic retinopathy (Front Immunol. 2020;11:583687. doi: 10.3389/fimmu.2020.583687. eCollection 2020) and diabetic nephropathy (Postgrad Med. 2023;135(5):519-523. doi: 10.1080/00325481.2023.2214058) are characterized with high burden of inflammation. Thus, early detection of diabetes and its complications could improve effective management of the disease and its complications.

Moreover, discussion must be improved by discussing similar works. Some examples are https://doi.org/10.1007/s13410-021-01040-5 and https://doi.org/10.1007/s13410-022-01136-6

Reviewer #3: (No Response)

7. PLOS authors have the option to publish the peer review history of their article (what does this mean?). If published, this will include your full peer review and any attached files.

Reviewer #1: No

Reviewer #3: No

---

## [Author Response · Author response to Decision Letter 1]

27 Dec 2023

Dear Reviewers,

We thank you for your thoughtful and constructive comments on our manuscript entitled " Investigation of the reasons for delayed presentation in proliferative diabetic retinopathy patients."

We have carefully considered your feedback and have made the following revisions to the manuscript.

In response to Reviewer #1's comments, we have added the following information to the introduction:

Inflammation is a common feature of type 2 diabetes and its complications, including diabetic retinopathy. (line 77)

Specifically, inflammation has been shown to be associated with the progression of diabetic neuropathy, diabetic retinopathy, and diabetic nephropathy. (line 78)

Early detection of diabetes and its complications, including diabetic retinopathy, could improve effective management of the disease and its complications. (line 80)

We have also added the following references to support these claims:

1. Swiss Med Wkly. 2019;149:w20139. doi: 10.4414/smw.2019.20139 (line 79)

2. Eur J Clin Invest. 2020;50(3):e13206. doi: 10.1111/eci.13206 (line 80)

3. Preprints 2023, 2023060202. doi:10.20944/preprints2023060202.v1 (line 79)

4. Front Immunol. 2020;11:583687. doi: 10.3389/fimmu.2020.583687. eCollection 2020 (line 80)

5. Postgrad Med. 2023;135(5):519-523. doi: 10.1080/00325481.2023.2214058 (line 80)

We have also revised the discussion section to address Reviewer #1's comments. Specifically, we have added the following information:

We have emphasized the potential role of inflammation in accelerating the progression of diabetic retinopathy in patients with poorly controlled diabetes. (line 482-485,506,554)

We have discussed the findings of previous studies that have shown a relationship between inflammation and the progression of diabetic retinopathy. (lines 482-485)

We have suggested that future studies should investigate the potential use of inflammation markers to identify patients at risk for progression of diabetic retinopathy. (lines 803-806)

In the methods and discussion limitations section, we have added the following sentence:

In line 163 and 806-807, we have added the following sentence to clarify that we did not include gestational diabetes mellitus patients in our study and therefore could not study the relationship between atrial conduction time and P-wave dispersion and DR progression.

We have not received any comments from Reviewer #3.

In addition to the specific revisions mentioned above, we have also made the following general revisions to the manuscript:

We have clarified the language and structure of the manuscript to improve readability.

We have corrected any errors in grammar or spelling.

We have added additional information to support the claims made in the manuscript.

We believe that these general revisions have improved the overall quality of the manuscript.

We believe that these revisions have addressed the concerns raised by you. We hope that you will find the revised manuscript to be satisfactory.

Sincerely,

Jipeng Li

Beijing Tongren Eye Center

---

## [Decision Letter · Decision Letter 2]

25 Jan 2024

Investigation of the reasons for delayed presentation in proliferative diabetic retinopathy patients

PONE-D-23-26728R2

Dear Dr. Li

We’re pleased to inform you that your manuscript has been judged scientifically suitable for publication and will be formally accepted for publication once it meets all outstanding technical requirements.

Kind regards,

Jiro Kogo

Academic Editor

PLOS ONE

Additional Editor Comments (optional):

Reviewers' comments:

Reviewer's Responses to Questions

**Comments to the Author**

1. If the authors have adequately addressed your comments raised in a previous round of review and you feel that this manuscript is now acceptable for publication, you may indicate that here to bypass the “Comments to the Author” section, enter your conflict of interest statement in the “Confidential to Editor” section, and submit your "Accept" recommendation.

Reviewer #3: All comments have been addressed

2. Is the manuscript technically sound, and do the data support the conclusions?

Reviewer #3: Yes

3. Has the statistical analysis been performed appropriately and rigorously? 

Reviewer #3: Yes

4. Have the authors made all data underlying the findings in their manuscript fully available?

Reviewer #3: Yes

5. Is the manuscript presented in an intelligible fashion and written in standard English?

Reviewer #3: Yes

6. Review Comments to the Author

Reviewer #3: The authors addressed the raised concerns from the reviewers, and the revised manuscript became much better.

7. PLOS authors have the option to publish the peer review history of their article (what does this mean?). If published, this will include your full peer review and any attached files.

Reviewer #3: No

---

## [Editor Report · Acceptance letter]

17 Feb 2024

PONE-D-23-26728R2 

PLOS ONE

Dear Dr. Li, 

I'm pleased to inform you that your manuscript has been deemed suitable for publication in PLOS ONE. Congratulations! Your manuscript is now being handed over to our production team.

Kind regards, 

on behalf of

Dr. Jiro Kogo 

Academic Editor

PLOS ONE